# Metabolic reprogramming in Rheumatoid Arthritis Synovial Fibroblasts: A hybrid modeling approach

**Sahar Aghakhani** [1,2], **Sylvain Soliman** [2], **Anna Niarakis** [1,2]*

**1** GenHotel–Laboratoire Européen de Recherche pour la Polyarthrite Rhumatoïde, Univ. Evry, Univ. Paris-Saclay, Evry, France, **2** Lifeware Group, Inria Saclay Île-de-France, Palaiseau, France

* anna.niaraki@univ-evry.fr

## Abstract

Rheumatoid Arthritis (RA) is an autoimmune disease characterized by a highly invasive pannus formation consisting mainly of Synovial Fibroblasts (RASFs). This pannus leads to cartilage, bone, and soft tissue destruction in the affected joint. RASFs' activation is associated with metabolic alterations resulting from dysregulation of extracellular signals' transduction and gene regulation. Deciphering the intricate mechanisms at the origin of this metabolic reprogramming may provide significant insight into RASFs' involvement in RA's pathogenesis and offer new therapeutic strategies. Qualitative and quantitative dynamic modeling can address some of these features, but hybrid models represent a real asset in their ability to span multiple layers of biological machinery. This work presents the first hybrid RASF model: the combination of a cell-specific qualitative regulatory network with a global metabolic network. The automated framework for hybrid modeling exploits the regulatory network's trap-spaces as additional constraints on the metabolic network. Subsequent flux balance analysis allows assessment of RASFs' regulatory outcomes' impact on their metabolic flux distribution. The hybrid RASF model reproduces the experimentally observed metabolic reprogramming induced by signaling and gene regulation in RASFs. Simulations also enable further hypotheses on the potential reverse Warburg effect in RA. RASFs may undergo metabolic reprogramming to turn into "metabolic factories", producing high levels of energy-rich fuels and nutrients for neighboring demanding cells through the crucial role of HIF1.

**Data Availability Statement:** To facilitate reproducibility, we compiled all analysis in python notebooks and R scripts, in a step-by-step annotated manner. All data and code used to generate results are available on a GitLab

## Author summary

We successfully built the first large-scale hybrid dynamical model for human Rheumatoid Arthritis Synovial Fibroblasts (RASFs) including signaling, gene regulation, and metabolism. We used a state-of-the-art molecular map for upstream signaling and gene regulation, the tool CaSQ to infer a large-scale Boolean model, and a human central metabolic model. Trap-spaces of the Boolean asynchronous model were used to infer additional metabolic constraints on the metabolic network for subsequent flux balance analysis. This method allowed us to study the impact of various regulatory initial conditions on RASFs' metabolic fluxes distribution. Our model successfully reproduces the metabolic

repository at https://gitlab.com/genhotel/rasf-hybrid-model and in a Zenodo permanent archive at https://doi.org/10.5281/zenodo.7181588.

**Funding:** The work was supported by the doctorate program of the University of Paris Saclay, France (S.A), Genopole (A.N and S.A) & INRIA (S.S and A.N). The funders had no role in study design, data collection and analysis, decision to publish, or preparation of the manuscript.

**Competing interests:** The authors have declared that no competing interests exist.

reprogramming of RASFs which shift their energy production from oxidative pathways to glycolysis, highlighting the key role of HIF1 in this process. Our findings allow us to hypothesize a reverse Warburg relationship occurring between RASFs and other RA joint cells. Similarly to Cancer-Associated Fibroblasts, RASFs would undergo a metabolic switch and reprogram their metabolism to adapt to their hypoxic environment and provide crucial metabolic intermediates to neighboring cells to sustain their inflammatory activity.

## 1. Introduction

Rheumatoid Arthritis (RA) is a chronic auto-immune disorder affecting approximately 0.5–1% of industrialized countries' population [1]. The growing proportion of people affected [2] and the impact on the patients' quality of life have turned RA into a severe public health issue. Resulting from the progressive destruction of cartilage and bones, RA leads to pain, swelling, and stiffness in the affected joint [1]. Extra-articular manifestations often follow, such as cardiac, renal or neurological disorders, all associated with excess mortality [3].

Although it is widely acknowledged that an immune system's dysfunction triggers RA, the etiology of the disease is not yet fully elucidated. RA is a complex disease resulting from multiple intertwined factors [4]. Genetic factors of susceptibility were initially identified [5], along with epigenetic factors, namely DNA methylation or histone proteins modifications [6]. Environmental factors have also been recognized recently, including smoking [7] and pollution [8]. Finally, RA's female-to-male prevalence ratio having consistently been established at 3:1 [9], hormonal factors are suspected to play a role [10].

This complexity along with the fragmented knowledge of the disease pathophysiology contribute to the lack of a cure for RA. The different treatment options aim to reduce further damage and symptoms to improve the patients' overall quality of life. Disease-modifying antirheumatic drugs (DMARDs), including hydroxychloroquine or methotrexate, may be used to promote remission by slowing or stopping the progression of the disease [11]. Nonsteroidal anti-inflammatory drugs (NSAIDs) such as acetylsalicylate, naproxen, and ibuprofen may also be considered to relieve pain and decrease inflammation [12]. A new lead lies in the action of JAK inhibitors, namely tofacitinib, to remedy inflammation [13]. However, despite the various options available, a significant proportion of RA patients do not respond significantly to treatment and are in a state of therapeutic distress [14].

RA's pathogenesis is associated with synovial hyperplasia or pannus formation, consisting of the accumulation of macrophages and synovial fibroblasts (RASFs), leading to inflammation and bone erosion [15]. RASFs are the most common resident cells of the synovial membrane and play a crucial role in the onset and disease progression [16]. They are derived from the joints' synovial membrane and differ from healthy fibroblasts' morphology, gene expression pattern, and phenotype. In a healthy joint, fibroblasts are arranged as a one or two-cell layer and are responsible for the synovial membrane's structural integrity [17]. In addition, fibroblasts ensure nutrient supply, joint lubrication, and create the synovial fluid's non-rigid extracellular matrix, aiding wound healing and tissue repair. On the contrary, the rheumatic pannus is a 10–15 cell depth layer [18] where RASFs present an aggressive tumor-like phenotype. They express very high levels of cytokines, chemokines, and matrix-degrading enzymes, causing cartilage damage and maintaining inflammation. Their reduced contact inhibition and their resistance to apoptosis allow them to proliferate massively, invading periarticular tissues and contributing to their destruction. In addition, RASFs are considered as primary

drivers of angiogenesis in RA joints [16,19]. In light of these advances, RASFs, once considered as passive bystanders, are now recognized as active players in disease pathogenesis. Therefore, RASF-directed therapies could become a complementary approach to currently used immune therapies [20].

Healthy fibroblasts' transformation into RASFs appears to be associated with metabolic alterations [21]. Under normoxic conditions, healthy cells oxidize one glucose molecule into two pyruvate molecules along with 36 molecules of adenosine triphosphate (ATP) through glycolysis followed by tricarboxylic acid cycle (TCA) and oxidative phosphorylation (OXPHOS). In hypoxic conditions, pyruvate is diverted from TCA and transformed into lactate, generating two molecules of ATP. In cancer cells, this metabolic switch can also occur in the presence of oxygen, known as the Warburg effect [22]. Cancer cells opt for glycolysis to produce energy in the form of ATP even if it is less efficient because glycolytic intermediates will feed metabolic pathways supporting cell growth, proliferation, and survival (e.g. pentose phosphate pathway (PPP), fatty acids, glutaminolysis). A few recent experimental studies indicated a link between altered metabolism and inflammation levels in RA. Measurements of synovial tissue $PO_2$ levels demonstrated that the RA joint microenvironment is profoundly hypoxic [23]. In addition, increased glucose uptake, glycolytic enzymes, lactate secretion, and oxidative damage have been identified in RASFs, correlating with cytokine levels and disease scores [24].

While provided with some understanding of the metabolic pathways altered in RASFs, there is still a lack of insight into their activating stimuli and the relevance of such metabolic reprogramming in the rheumatic joint. As fragmented as it may be, a collection of the knowledge in terms of extra- and intra-cellular signaling along with gene regulation pathways involved in RASFs' metabolic reprogramming would be beneficial. Efforts have already been made in this direction with the publication of the first RA-map [25] followed by the RA-Atlas [26]. The RA-Atlas is an interactive, manually curated representation of molecular mechanisms involved in RA's pathogenesis following Systems Biology Graphical Notation standards [27]. It includes an updated version of the RA-map, the RA-map V2, with the addition of metabolic machinery, and cell-specific molecular interaction maps for CD4+ Th1 cells, fibroblasts, M1 and M2 macrophages. Although including the impact of signaling and gene regulation on four major metabolic pathways, namely glycolysis, PPP, TCA, and OXPHOS, the RA-map V2 is limited by its non-cellular specificity and static knowledge base function. However, it can be used to construct dynamic computational models to extend knowledge with executable information.

Dynamic modeling is necessary to understand the emergent behavior of biological entities when complex and intertwined pathways are involved. It helps elucidate complex mechanisms occurring at different scales (e.g. signaling, gene regulation, and metabolism) between different entities (e.g. ligands, receptors, proteins, metabolites, genes, RNAs). While dynamic models based on quantitative information provide great insight into biological systems, qualitative models based on logical relationships among components provide an appropriate description for systems with unknown mechanistic foundations or lacking precise quantitative data [28]. Qualitative Boolean models allow the parameter-free study of large-size biological pathways' underlying dynamic properties. In Boolean formalism, nodes represent regulatory components and arcs represent their interactions. Each regulatory component is associated with a Boolean value (0 or 1), indicating its qualitative concentration (absent or present) or its activity level (inactive or active). The state of each node depends on the state of its upstream regulators and is described by a Boolean rule defined by the logical operators "AND", "OR" and "NOT". Despite being well-suited to account for the modeling of signaling and gene regulation mechanisms, qualitative Boolean modeling is not appropriate to assess quantitative metabolic properties. Where signaling and gene regulation carry signal flow, metabolism generates mass flow. A

widely used method for analyzing metabolic networks is Flux Balance Analysis (FBA). The latter is a mathematical method used in large-scale reconstructions of metabolic networks [29]. Its main advantage lies in the need for little information regarding enzymes' kinetic parameters and metabolites' concentrations as it calculates the flow of metabolites by assuming steady state conditions.

As outlined above, the various biological features of signaling, gene regulation, and metabolism are usually studied separately through different modeling formalisms. However, considering that a biological phenotype results from their interoperation, a more hybrid formalism capable of spanning these layers would be highly beneficial in studying complex diseases [30–32]. One of the recent efforts in this direction is the FlexFlux framework [33] combining FBA and qualitative simulations by seeking regulatory steady states through synchronous updates of multi-state qualitative initial values. Van der Zee & Barberis also proposed a framework to integrate Boolean with constraint-based models of metabolism [34].

Hybrid modeling has already been suggested in RA to improve understanding of the disease, notably with a framework describing pannus formation [35], but never in a cell-specific manner. In this regard, this work presents the first hybrid RASF model. It was obtained by combining a cell-specific asynchronous Boolean network covering gene regulation and signaling machinery with a global constraint-based metabolic network. This hybrid model allows bridging the gap between various biological features to assess the impact of RASFs' regulatory outcomes on their central metabolic flux distribution.

## 2. Methods

### 2.1. RASF regulatory model

**2.1.1. Inference of a Boolean model from the RA-map V2.** The RA-map V2, included in the RA-Atlas [26], provided the starting point for this work. It is the largest, fully annotated, mechanistic representation of existing knowledge related to the onset and progression of RA. It illustrates the central signaling, gene regulation and metabolic pathways (i.e. glycolysis, PPP, TCA, and OXPHOS), along with molecular mechanisms and phenotypes involved in RA's pathogenesis. The RA-map V2 is based on the manual curation of 575 peer-reviewed scientific publications and comprises 720 components and 602 reactions. It is an upgrade of the original global RA-map [25] with the addition of metabolic information and enrichment of signaling and regulatory pathways in RASF-specific knowledge. The latter were added from manual curation of omics experiments-associated publications. Additionally, the manual curation process allowed for disease and cell specificity elevating the confidence of the included information. Interactions depicted in the RA-map V2 are not protein-protein interactions, inferred from interaction databases, but mechanistic, causal interactions based on experimental evidence. For more information about the RA-map V2 construction, please refer to [25,26]. Despite the RASF-specific enrichment, the RA-map V2 remains a global map, gathering information from several cell types, tissues, and fluids such as RASFs, synovial tissue, synovial fluid, blood and serum components, peripheral blood mononuclear cells (PBMC), chondrocytes, and macrophages. CaSQ [36] was used to infer a Boolean model from the RA-map V2 in the standard Systems Biology Marked up Language-qualitative (SBML-*qual*) [37] format. Its latest functionality enables extraction and translation into a model, not an entire molecular map, but only a subpart of interest. This feature allowed overcoming the RA-map V2's lack of cellular-specificity by only extracting pathways upstream and downstream of RASF-specific extracellular ligands, metabolites, and microRNA experimentally demonstrated to be significant in RA's pathogenesis.

**2.1.2. Cellular-specificity assessment.**   The cellular-specificity of the model was evaluated by comparing its components' cellular origin to eight different sample-type specific lists provided in the RA-Atlas publication [26]. Lists are specific to RASFs, macrophages, synovial tissue, synovial fluid, blood and serum components, PBMC, and chondrocytes. First two are obtained by aggregating single-cell omics data and literature data. Remaining sample-type specific lists are obtained through literature mining. Regarding RASFs, said omics data include differentially expressed genes from GSE109449 [38], a RASF-specific single cell RNA-seq analysis dataset.

**2.1.3. Annotation score calculation.**   Thorough bibliographic annotations of the RA-map V2 in the form of PubMed IDs (PMIDs) following MIRIAM (Minimum Information Required In The Annotation of Models) standards [39], were kept in the associated model. They allowed to calculate annotation scores for each compound present in the model based on the number of bibliographic references describing it.

**2.1.4. Validation of the regulatory model's behavior.**   The model's behavior was assessed to confirm its biological relevance. Simulations were performed on Cell Collective [40], an interactive platform to simulate and analyze biological models. Experimental evidence for RASF-specific scenarios were retrieved from *in-vitro* and *in-vivo* studies, in humans when possible, but mostly in murine models of RA. Experimental conditions were used as the model's initial conditions, and its outputs were compared to the biological outcome to confirm or refute a scenario's validation. Two approaches were carried out to assess the model's behavior. First, a generic validation was performed to verify specific compounds' contribution to signaling or gene regulation pathways. This mechanistic verification was performed in a synthetic state of the model where all nodes' values were set to 0 (i.e. absent/inactive) and specific compounds to test were set alternatively to 1 and 0. Secondly, the global behavior of the model was analyzed under RASF-specific initial conditions extracted from literature. Initial values are assigned to all regulatory model's inputs (i.e. extracellular ligands which do not have any upstream regulators) and a few intermediate nodes to reproduce RASF-specific conditions. Inputs are suspected to exert a significant control on the model's dynamics due to the linearity of signal transduction in the RA-map V2 and associated model. This global analysis, conducted at the level of the model's phenotypes, allowed to evaluate the different pathways' interconnection and compare the global model's behavior to RASFs' experimentally expected behavior. In both cases, simulations were performed in the asynchronous updating mode, with a simulation speed of one and a sliding window of 30.

## 2.2. Metabolic model

The metabolic network used in this work is the MitoCore model [41], a manually curated constraint-based model of human central metabolism. It includes two compartments (i.e. cytosol and mitochondria), 74 metabolites, 324 metabolic reactions, and 83 transport reactions and covers all parts of central metabolism directly or indirectly involved with energy production. This "core" model, although smaller in size compared to recently published human genome-scale metabolic models, allows to avoid many large genome-scale models' associated issues (e.g. unrealistic ATP production rates, automatic and not-curated reconstruction of improper gene-protein-reactions rules leading to incorrect compartmentalization of reactions or directionality constraints). Additionally, considering its manual curation, users can have great confidence in each reaction and have a better insight on the system's behavior, allowing for an easier evaluation of the results. The default parameters of MitoCore simulate normal cardiomyocyte metabolism. However, the default simulation settings can be applied to various biological contexts without necessarily implying cell-specific features. Indeed, cardiomyocytes

can metabolize a wide range of substrates, have reactions common to many other cell types, and represent the metabolism of the human heart, an organ of utmost importance in human health, disease, and toxicology. Moreover, MitoCore includes processes that are inactive in cardiomyocytes but that can be activated to represent other cell-type's metabolic features (e.g. gluconeogenesis, ketogenesis, β-alanine synthesis and folate degradation). As a proof of concept regarding the generalization of their model, MitoCore modelers were able to successfully simulate the fumarase deficiency, a nervous system condition, using the default cardiomyocyte parameters.

### 2.3. Framework for hybrid modeling

The general architecture of the framework for hybrid modeling is provided in Fig 1.

**2.3.1. Value propagation.** Value propagation [42,43], a method implemented in the CoLoMoTo notebook [44], was applied to the qualitative regulatory model to facilitate its analysis. When given a set of logical rules and a cellular context, this iterative algorithm allows the computation of specific components' dynamical consequences on the model's behavior. It reveals the influence specific compounds may exert on the network's dynamics. Note that this method does not impact the asymptotic behavior of the model: all dynamical consequences calculated in this manner would occur regardless.

**2.3.2. Identification of regulatory trap-spaces.** Evaluating the influence of RASF-specific components on the global model's behavior through value propagation allows to decrease its complexity to identify trap-spaces. Trap-spaces, also called stable motifs or steady symbolic states, are parts of the dynamics from which the system cannot escape [45,46]. Trap-spaces may overlap or include each other. Thus, minimal trap-spaces (later referred to as "trap-spaces" for readability), i.e. trap-spaces which do not include smaller trap-spaces, offer a good approximation of attractors and faithfully capture the asymptotic behavior of Boolean models.

The outputs of value propagation were considered as a new set of initial conditions and the biolqm.trapspace function was used to identify the model's trap-spaces. Their computation relies on the identification of positive and negative prime implicants for each component's function without performing simulation but rather through a symbolic approach implementing a constraint-solving method [45,46]. In bioLQM [47], trap-spaces can be obtained directly using binary decision diagrams or an ASP-based solver.

**2.3.3. Projection of metabolic trap-spaces.** The regulatory model does not allow to differentiate a protein with a signaling function from a metabolic enzyme or any simple molecule from a metabolite. To overcome this limitation, all regulatory model's components were extracted as a list, as well as MitoCore's enzymes and MitoCore's metabolites. Both regulatory and metabolic network's metabolic components being consistently named with BiGG IDs [48], the regulatory model's components were compared to MitoCore's metabolites and a list of the regulatory model's metabolites was obtained. This matching was limited by excluding a list of manually predefined compounds considered by MitoCore as metabolites but actually are common metabolic intermediates (S1 Table). Similarly, a list of the regulatory model's enzymes was extracted. Said lists were used to project previously identified regulatory trap-spaces on the metabolic enzymes and metabolites. The latter reflected RASFs' signaling and gene regulation's impact on cellular metabolism, i.e. cellular phenotype.

**2.3.4. Constraining metabolic fluxes.** The maximal value of trap-spaces relative to metabolic compounds were used to constrain associated metabolic flux. A metabolic enzyme-associated maximal trap-space value strictly superior to 0 means this metabolic enzyme might

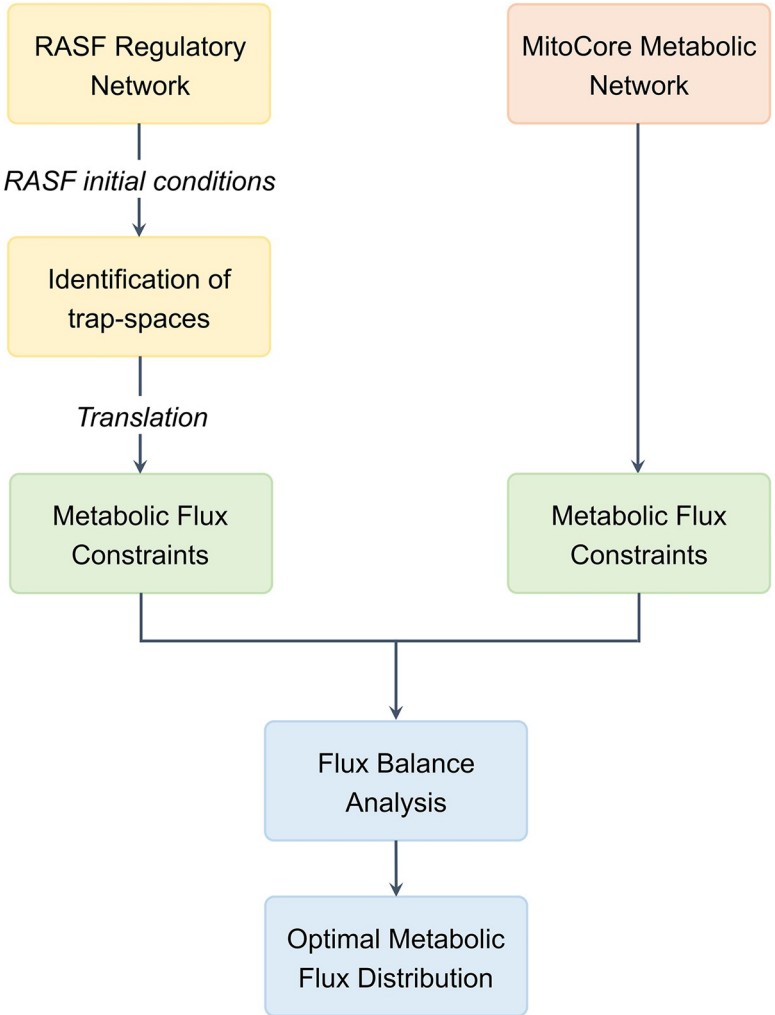

**Fig 1. General architecture of the hybrid modeling framework.**

be activated or present according to the regulatory model's outcome. An enzyme being qualitatively activated or present does not give information about the feasibility nor the kinetics of the metabolic reactions it catalyzes. Similarly, a metabolite-relative maximal trap-space value strictly superior to 0 shows the metabolite is produced in some of these regulatory conditions. However, it does not give information about its producing reactions' high or low flux. Thus, it is impossible to influence the reaction flux of metabolic compounds associated with maximal trap-spaces values strictly superior to 0.

On the other hand, a metabolic enzyme with a projected maximal trap-space value equal to 0 (i.e. trap-spaces always equal to 0) expresses the inactivation or absence of the said enzyme by signaling or gene regulation pathways. The catalyzed metabolic reactions will not happen. A metabolite-associated maximal trap-space value of 0 denotes the non-production of the said metabolite. Its producing reactions will not occur. Hence, for every metabolic enzyme or metabolite with projected maximal regulatory trap-spaces values equal to 0, the flux of its associated MitoCore metabolic reaction (i.e. catalyzed reactions or producing reactions) is constrained to 0.

## 2.4. Metabolic network analysis

**2.4.1. Flux Balance Analysis.** Flux Balance Analysis (FBA) was performed using CobraPy [49] to evaluate MitoCore's metabolic flux distribution. Two FBAs were conducted to highlight a potential change in metabolic fluxes' distribution under RASF-specific conditions. The first FBA was conducted without additional constraints, reflecting the healthy control state. The second FBA was conducted with additional metabolic flux constraints extracted from the regulatory model's trap-spaces, reflecting RASFs' condition. The objective function was set to maximum cellular ATP production to reflect the primary energy production role of central metabolism. It was manually defined as the sum of the three cellular ATP-producing reactions (i.e. the seventh and tenth reactions of glycolysis, and OXPHOS' last step).

The MitoCore metabolic model lacking cellular and tissular specificity, a numerical interpretation of metabolic flux values was not possible. FBA results were only interpreted in terms of metabolic flux distribution. For instance, the ratios of ATP production from glycolytic or OXPHOS reactions relative to total ATP production (represented by the objective function) were calculated. In addition, an analysis of uptake and secretion of carbon fluxes (C-flux) was carried out to indicate the central carbon metabolism. The C-flux of an uptake reaction represents the total cellular carbon influx from this specific uptake reaction. Similarly, the C-flux of a secretion reaction represents the proportion of total cellular carbon efflux coming from this specific secretion reaction. Finally, a comparison of both FBAs' internal fluxes was conducted and a difference in metabolic fluxes was identified from a greater than 2-fold variation.

**2.4.2. Regulatory initial condition's knock-out/knock-in simulations.** To identify regulatory pathways potentially responsible for RASF's metabolic alterations, a knock-out/knock-in strategy of the regulatory network's initial conditions was conducted. All initial regulatory conditions set to 1 in the RASF-specific configuration were successively set to 0 while the other remained at RASF-specific values. Similarly, all regulatory components' initial conditions set to 0 in the RASF-specific configuration were successively set to 1 while the other remained at RASF-specific values. Subsequent value propagation, metabolic compounds-projected regulatory trap-spaces' identification and extraction of metabolic constraints were conducted to perform FBA and evaluate metabolic fluxes' distribution in these new conditions. To depict the metabolism's key function of energy production, the proportion of total cellular ATP production from glycolysis and OXPHOS were compared in the various FBAs.

# 3. Results

## 3.1. RASF regulatory model

Translation of the RA-map V2 by focusing on RASF-specific molecular pathways (Table 1) provided a dynamic Boolean model of 359 nodes, including 14 inputs, and 642 interactions (S1 Fig).

Although the RA-map V2 is a collection of information from several cell types, tissues and fluids, the inferred model is mostly RASF-specific (82%) (Fig 2A). When interpreting those results, one must consider that a specific component can be common to several cell types. If only exclusive components are considered, the model is 91% RASF-specific (Fig 2B). This high level of cellular specificity, along with the phenotypes and extracellular ligands specificity, allows referring to this model as the RASF model.

In addition, the RASF model's annotation score enables high confidence in its knowledge base function. Indeed, 97% of the model's components present more than one manually curated experimental evidence and 74% present more than 2 (Fig 2C). Various situations can be considered regarding the 3% of compounds for which no bibliographic reference is given.

**Table 1. RASF-specific components and the export direction used to infer a Boolean model from the RA-map V2 using CaSQ.**

| Export direction | Component | Class | Reference |
|---|---|---|---|
| Upstream | HIF1 | Protein | [50] |
| Downstream | FASLG | Extracellular ligand | [51] |
| Downstream | FGF1 | Extracellular ligand | [52] |
| Downstream | GLC | Extracellular metabolite | [53] |
| Downstream | IKBA/NFKB1/RELA | Extracellular complex | [54] |
| Downstream | IL17A | Extracellular ligand | [55] |
| Downstream | IL18 | Extracellular ligand | [56] |
| Downstream | IL6 | Extracellular ligand | [57] |
| Downstream | MIR192 | microRNA | [58] |
| Downstream | PDGFA | Extracellular ligand | [59] |
| Downstream | RANKL | Extracellular ligand | [60] |
| Downstream | SFRP5 | Extracellular ligand | [61] |
| Downstream | TGFB1 | Extracellular ligand | [62] |
| Downstream | TNF | Extracellular ligand | [63] |
| Downstream | WNT5A | Extracellular ligand | [64] |

Many are simple molecules acting as products or reactants of well-known biological reactions whose expressions are rarely highlighted in disease-specific experimental studies (e.g. ATP, ADP, NADH, NADPH, H2O, O2, FADH, Ca2+). A further distinction is made regarding pathway intermediates. For instance, a specific pathway might be experimentally proven to be expressed in a disease-specific manner, but not all intermediates are necessarily studied. It is the case for several metabolic pathways that are found to be expressed in RASFs but experimental evidence is not available for every component.

To validate the RASF-specific behavior of the model, generic RASF model simulations were first compared to experimental scenarios (S2 Table). Regarding this evaluation, 23 experimental scenarios were confirmed out of 30. The scenarios that were not reproducible were due to multiple reasons. First, mechanistic information regarding certain interactions may be lacking in the literature, leading to a missing or incomplete representation in the RA-map V2 and the associated RASF model. Additionally, the validation of some scenarios involved stoichiometric information that was not possible to reproduce with a strictly Boolean formalism. Finally, some generic scenarios were not validated since other pathways already activated/inactivated the said phenotype in the same initial conditions. Subsequent global model's behavior evaluation under RASF-specific initial conditions (Table 2) confirmed RASFs' experimental behavior. Indeed, aggressive phenotypes, i.e. bone erosion, cell chemotaxis, cell growth, hypoxia, inflammation, matrix degradation, and osteoclastogenesis, are ON while apoptosis is OFF (Fig 3).

### 3.2. Framework for hybrid modeling

**3.2.1 Identification of regulatory trap-spaces.** The value propagation method was applied to the regulatory network under RASF-specific initial conditions (Table 2). Out of the 359 components present in the RASF model, 313 were fixed by value propagation (100 were fixed at 0 and 213 at 1). Evidently, the RASF-specific initial conditions, including 14 inputs and 2 intermediary nodes, exert an important control over the whole network.

Using the results of value propagation as a new set of initial conditions enabled to decrease the RASF model's complexity to obtain its trap-spaces, including the complete asymptotic behavior of the system (S3 Table). Eight different trap-spaces were identified, each trap-space reflecting a different subspace of RASFs' cellular phenotypes. Most components' values are

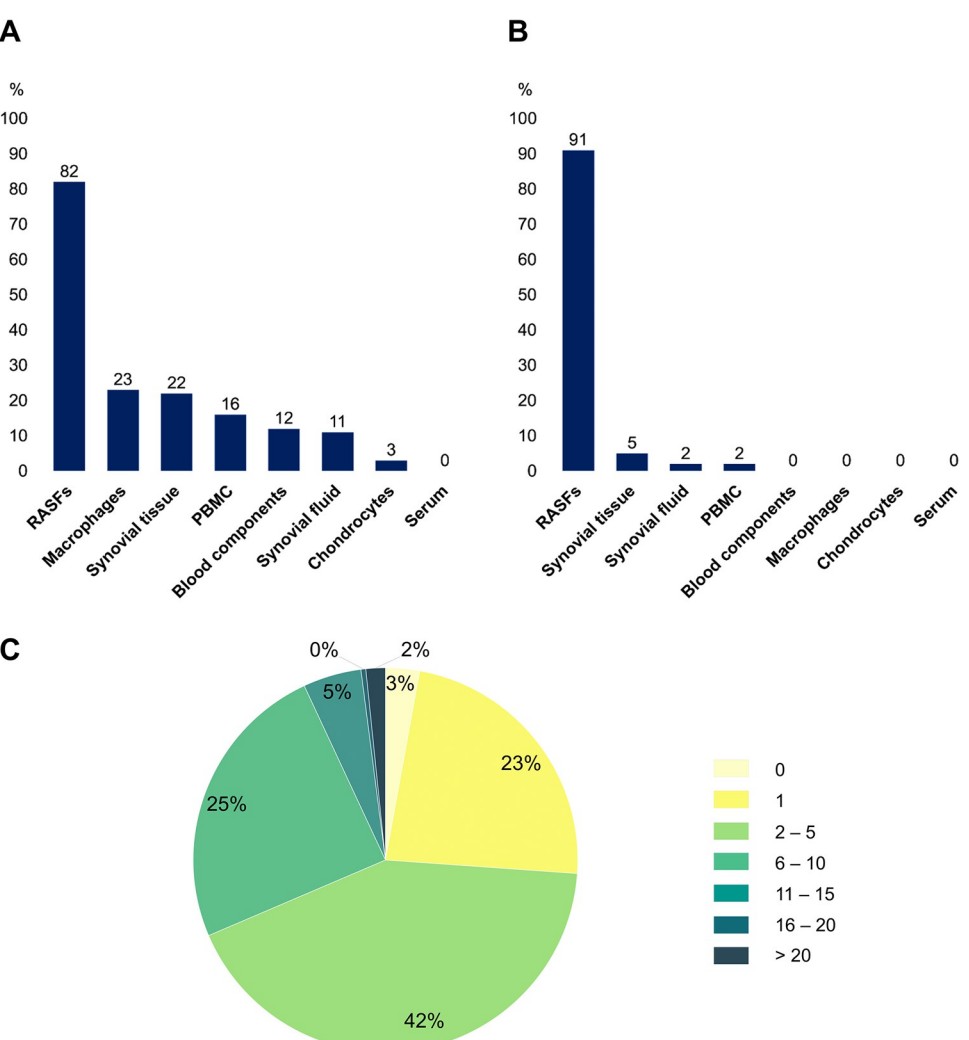

**Fig 2.** Statistical analysis of the RASF model. **(A)** Distribution of the RASF model's cellular specificity. **(B)** Distribution of the RASF model's cellular specificity when only exclusive components are considered. **(C)** Annotation score's distribution among the RASF model's components.

stable within all trap-spaces (always fixed either at 0 or 1) but others vary within trap-spaces. If we restrict the projection of the trap-spaces to the nine RASF model's ontological phenotypes, we obtain the results presented in Table 3. Ontological phenotypes refer to nodes representing a distinct cellular outcome in the RASF model.

As observed in Table 3, the phenotypes of angiogenesis, bone erosion, cell chemotaxis, cell growth, inflammation, matrix degradation and osteoclastogenesis exhibit an asymptotic stable active state when the model is simulated under RASF-specific conditions. These results are consistent with known RASFs' biological aggressive behavior as described in scientific literature [65].

The asymptotic state of ontological phenotypes is the result of the combined regulation exerted by their upstream regulators, as described in the logical formulas. Thus, the behavior of biomarker groups associated with the ontological phenotypes can be identified in the different trap-spaces. Subsequently, this behavior can be used for comparison against experimental evidence.

**Table 2. RASF-specific initial conditions extracted from literature.** Regulatory model's inputs are marked with an asterisk.

| Component | RASF-behavior | Source | Value |
|---|---|---|---|
| HIF1 | Activated | [50] | 1 |
| FASLG* | Activated | [51] | 1 |
| FGF1* | Activated | [52] | 1 |
| GLC* | Present | [53] | 1 |
| ATP | Present | [53] | 1 |
| IKBA/NFKB1/RELA* | Activated | [54] | 1 |
| IL17A* | Activated | [55] | 1 |
| IL18* | Activated | [56] | 1 |
| IL6* | Activated | [57] | 1 |
| MIR192* | Down-regulated | [58] | 0 |
| PDGFA* | Activated | [59] | 1 |
| RANKL* | Activated | [60] | 1 |
| SFRP5* | Down-regulated | [61] | 0 |
| TGFB1* | Activated | [62] | 1 |
| TNF* | Activated | [63] | 1 |
| WNT5A* | Activated | [64] | 1 |

For instance, all interleukins (e.g. IL121, IL18, IL1B, IL33, IL6) being active under RASF-specific conditions accounts for the asymptotic active state of the ontological inflammatory phenotype and confirms their experimentally observed function of inflammation drivers [66].

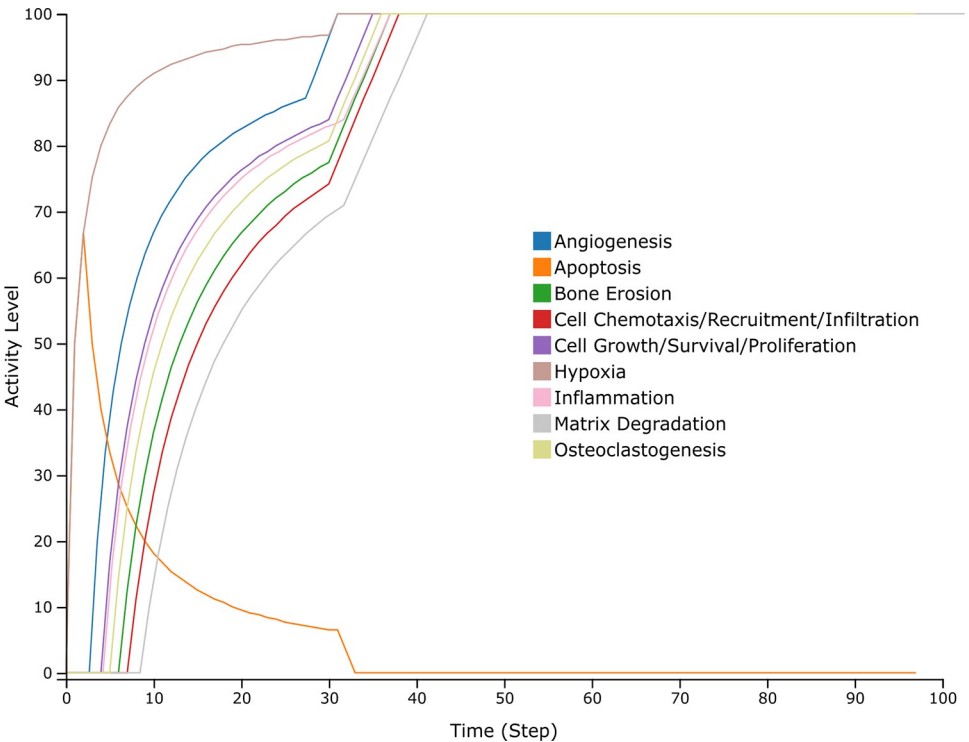

**Fig 3. Simulation of the RASF model under RASF-specific initial conditions performed on the Cell Collective platform.** The different curves depict the state of the phenotype components.

**Table 3. Projection of the RASF model's trap-spaces on its nine ontological phenotypes.**

| Ontological phenotype | Trap-space | | | | | | | |
|---|---|---|---|---|---|---|---|---|
| | 0 | 1 | 2 | 3 | 4 | 5 | 6 | 7 |
| Angiogenesis | 1 | 1 | 1 | 1 | 1 | 1 | 1 | 1 |
| Apoptosis | 0 | 0 | 0 | 0 | 0 | 0 | 0 | 0 |
| Bone Erosion | 1 | 1 | 1 | 1 | 1 | 1 | 1 | 1 |
| Cell Chemotaxis, Recruitment, Infiltration | 1 | 1 | 1 | 1 | 1 | 1 | 1 | 1 |
| Cell Growth, Survival, Proliferation | 1 | 1 | 1 | 1 | 1 | 1 | 1 | 1 |
| Hypoxia | 1 | 0 | 1 | 0 | 0 | 0 | 1 | 1 |
| Inflammation | 1 | 1 | 1 | 1 | 1 | 1 | 1 | 1 |
| Matrix Degradation | 1 | 1 | 1 | 1 | 1 | 1 | 1 | 1 |
| Osteoclastogenesis | 1 | 1 | 1 | 1 | 1 | 1 | 1 | 1 |

Likewise for the majority of the matrix metalloproteinases (e.g. MMP3, MMP9, MMP13) leading to matrix degradation [67] or cytokines (e.g. TNF, IL17) activating bone erosion and osteoclastogenesis [68].

A pattern of growth factors' (e.g. PDGFA, FGF1, VEGFA) activation is observed within the eight different trap-spaces and is associated with activation of the cell growth and proliferation phenotype. These findings are consistent with experimental evidence indicating them as key factors [69].

This proliferative behavior is confirmed in parallel by an asymptotic inactive state of the apoptotic phenotype, reproducing fibroblasts' resistance to programmed cell death in RA [70]. Accordingly with biological knowledge, it is due to the active state of anti-apoptotic components (e.g. CAV1) and the inactivity of pro-apoptotic ones (e.g. Bak, Bax) in all eight trap-spaces under RASF-specific conditions.

Finally, the hypoxic phenotype varies within trap-spaces, reflecting a biologically relevant feed-back loop. Trap-spaces where hypoxia is active are associated with active HIF1 and inactive PHD2. On the contrary, trap-spaces where hypoxia is inactivated are associated with inactive HIF1 and active PHD2. It reflects the well-known regulation of the cellular response to hypoxia by PHD2 through HIF1 [71].

The variations of fixed values within the eight trap-spaces can also be interpreted at the level of the RASF model's regulatory pathways. For instance, in trap-spaces 0, 1, 5, and 7, MAPK1 is active in parallel with BCL2. The latter are inactive in trap-spaces 2, 3, 4, and 6. It reflects the regulation of BCL2 through the MAPKs pathway [72].

**3.2.2 Projection of metabolic trap-spaces and constraints extraction.** Within the hybrid modeling framework, only metabolic components with proven inactive states are considered to extract additional metabolic constraints (S2 Fig). Using RASF-specific regulatory conditions, the maximal trap-spaces associated with seven metabolic enzymes and 12 metabolites were equal to 0. According to the hybrid modeling framework, this led to constrain 52 Mito-Core's metabolic reaction fluxes to 0. For instance, the maximal trap-space value relative to the metabolic enzyme AKGDm (2-Oxoglutarate Dehydrogenase) was equal to 0. Thus, the metabolic flux of the reaction it catalyzes (i.e. R_AKGDm) was constrained to 0 in MitoCore. Similarly, the maximal trap-space value associated with the metabolite fum_m (mitochondrial fumarate) was equal to 0. Consequently, its two metabolic producing reactions (i.e. R_CII_MitoCore and R_FUMtmB_Mitocore) were constrained to 0 in MitoCore. Interestingly, several metabolic reactions' fluxes (e.g. R_AKGDm, R_ICDHxm, R_CII_MitoCore, R_PDHm) were constrained to 0 in two distinct ways. Said reactions were constrained to 0 both by the extraction of metabolic constraints from the regulatory trap-spaces of the enzyme catalyzing the said

reaction and from the metabolite produced by the said reaction. Such dually extracted constraints reflect the consistency of RASFs' signaling and gene regulation's impact on their metabolic pathways. For a detailed list of metabolic enzymes and metabolites with maximal regulatory trap-spaces values equal to 0 and the associated constrained reactions in MitoCore, refer to S4 and S5 Tables.

## 3.3. Metabolic network analysis

A first FBA was carried out to enable metabolic flux distribution's assessment in a control situation. In addition, it allowed comparison with the second FBA, including metabolic constraints extracted from the RASF regulatory model. Results of both FBAs can be visualized in Fig 4A and 4B.

As expected biologically, the optimal fluxes for maximum cellular ATP production in a control situation are TCA and OXPHOS fluxes. They are responsible for 96% of total ATP production. In a control situation, the main uptaken carbonated molecules are hexadecanoate (40.92% of total C-flux), glucose (29.48%), lactate (9.42%), and $HCO_3$ (9.34%), primary energy

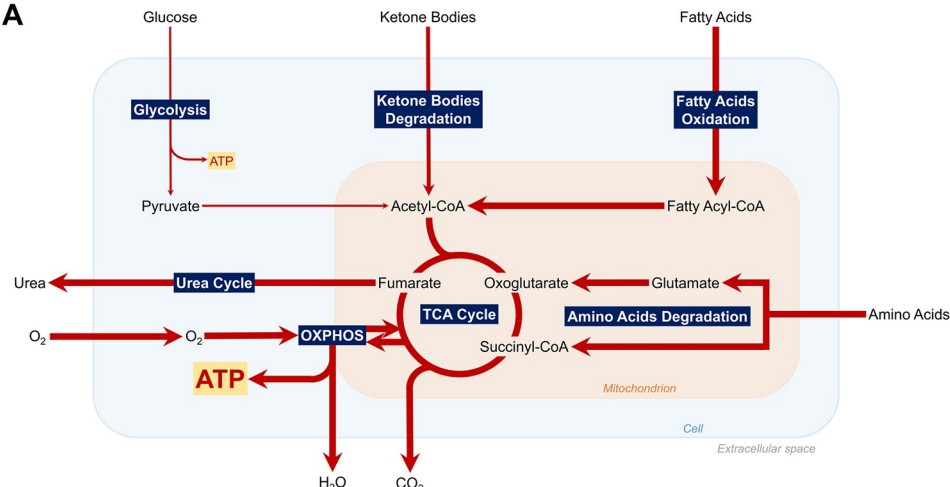

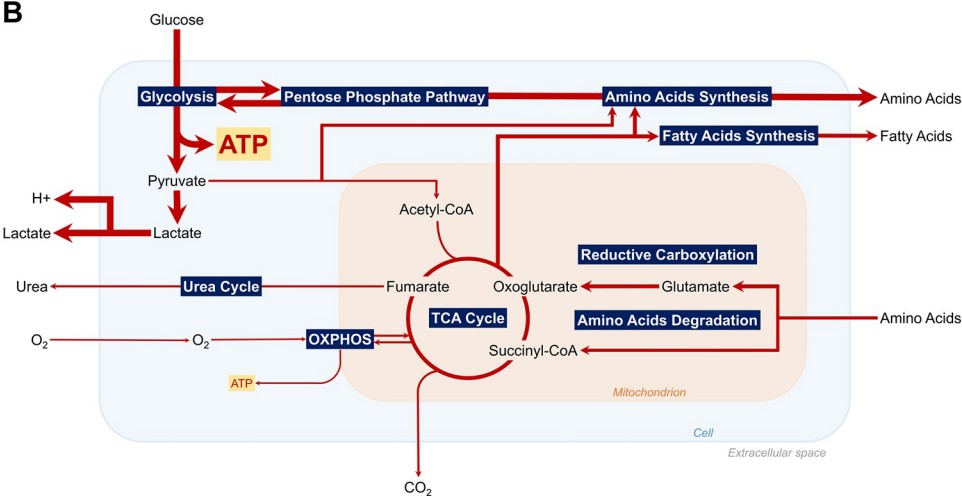

**Fig 4.** Summary of the major active pathways of central metabolism according to Flux Balance Analysis in control **(A)** and RASF-specific conditions **(B)** with maximum cellular ATP production as objective function.

sources for most cells. The main secreted carbonated molecule is $CO_2$ (99.96%) as it is the principal product of oxidative metabolism along with $H_2O$. In RASF-specific regulatory conditions, the optimal fluxes for maximum cellular ATP production are glycolytic, accounting for 85% of cellular ATP production. Metabolic uptake and secretion fluxes are also affected. The main uptaken carbonated molecules are glucose (85.69% of total C-flux) and aspartate (9.77%). The main secreted carbonated molecules are lactate (86.12%) and alanine (8.33%). Details of the uptake and secretion C-fluxes results for both FBAs can be found in S6–S9 Tables.

In addition, a comparison of internal metabolic fluxes (S10 and S11 Tables) in both situations illustrates increased glycolytic fluxes along with increased glucose uptake and lactate secretion in RASFs, accounting for a highly glycolytic metabolism. Low oxidative metabolism is demonstrated through decreased (almost null) TCA and OXPHOS fluxes and decreased secretion of OXPHOS by-products such as $CO_2$ and $H_2O$. A hypoxic environment is displayed with decreased $O_2$ uptake and increased $H^+$ secretion associated with environment acidity. Beyond metabolic pathways of ATP production, results denote reprogramming of several other metabolic pathways in RASFs. An increased amino-acids and fatty acids' secretion is shown, potentially acting in RA as substrates for energy production, biosynthesis intermediates, components of membrane phospholipids, or support for bone erosion and cartilage degradation. Increased reductive carboxylation is also identified, a novel glutamine metabolism pathway supporting the growth of tumor-like cells with mitochondrial defects. Further pathways including mitochondrial transport reactions, cardiolipin synthesis, or glycine cleavage, appear to be impacted, most likely indirectly as a result of metabolites' redirection through up-regulated metabolic pathways.

To decipher the role of regulatory components in RASFs' metabolic alterations, 14 FBAs were conducted following different sets of initial regulatory conditions. As shown in Table 4, out of the 14 RASF-specific initial conditions variants, only condition 3 (C3) significantly impacted ATP production pathways. Indeed, when inhibiting Hypoxia-Inducible Factor 1

**Table 4. Regulatory knock-out/knock-in simulations' set of initial conditions along with their FBA results in terms of total cellular ATP production from glycolysis and OXPHOS.** Each component's value initially set to 1/0 in RASF-specific conditions was alternatively set to 0/1 while the others remained at RASF-specific values.

| Component | Set of Initial Conditions | | | | | | | | | | | | | |
|---|---|---|---|---|---|---|---|---|---|---|---|---|---|---|
| | C1 | C2 | C3 | C4 | C5 | C6 | C7 | C8 | C9 | C10 | C11 | C12 | C13 | C14 |
| FASLG | 0 | 1 | 1 | 1 | 1 | 1 | 1 | 1 | 1 | 1 | 1 | 1 | 1 | 1 |
| FGF1 | 1 | 0 | 1 | 1 | 1 | 1 | 1 | 1 | 1 | 1 | 1 | 1 | 1 | 1 |
| HIF1 | 1 | 1 | 0 | 1 | 1 | 1 | 1 | 1 | 1 | 1 | 1 | 1 | 1 | 1 |
| IKBA/NFKB1/RELA | 1 | 1 | 1 | 0 | 1 | 1 | 1 | 1 | 1 | 1 | 1 | 1 | 1 | 1 |
| IL17A | 1 | 1 | 1 | 1 | 0 | 1 | 1 | 1 | 1 | 1 | 1 | 1 | 1 | 1 |
| IL18 | 1 | 1 | 1 | 1 | 1 | 0 | 1 | 1 | 1 | 1 | 1 | 1 | 1 | 1 |
| IL6 | 1 | 1 | 1 | 1 | 1 | 1 | 0 | 1 | 1 | 1 | 1 | 1 | 1 | 1 |
| MIR192 | 0 | 0 | 0 | 0 | 0 | 0 | 0 | 1 | 0 | 0 | 0 | 0 | 0 | 0 |
| PDGFA | 1 | 1 | 1 | 1 | 1 | 1 | 1 | 1 | 0 | 1 | 1 | 1 | 1 | 1 |
| RANKL | 1 | 1 | 1 | 1 | 1 | 1 | 1 | 1 | 1 | 0 | 1 | 1 | 1 | 1 |
| SFRP5 | 0 | 0 | 0 | 0 | 0 | 0 | 0 | 0 | 0 | 0 | 1 | 0 | 0 | 0 |
| TGFB1 | 1 | 1 | 1 | 1 | 1 | 1 | 1 | 1 | 1 | 1 | 1 | 0 | 1 | 1 |
| TNF | 1 | 1 | 1 | 1 | 1 | 1 | 1 | 1 | 1 | 1 | 1 | 1 | 0 | 1 |
| WNT5A | 1 | 1 | 1 | 1 | 1 | 1 | 1 | 1 | 1 | 1 | 1 | 1 | 1 | 0 |
| **Glycolysis** | 85.1 | 85.1 | 0 | 85.1 | 85.1 | 85.1 | 85.1 | 85.1 | 85.1 | 85.1 | 85.1 | 85.1 | 85.1 | 85.1 |
| **OXPHOS** | 14.9 | 14.9 | 100 | 14.9 | 14.9 | 14.9 | 14.9 | 14.9 | 14.9 | 14.9 | 14.9 | 14.9 | 14.9 | 14.9 |

(HIF1) and keeping RASF-specific initial conditions for the remaining ones, glycolysis was dramatically decreased and OXPHOS explained the cellular ATP production. This situation, although extreme in its proportions due to the framework's constraint extraction rules, is closer to a control situation. This finding suggests that targeting HIF1 could participate in restoring a healthy metabolic profile in RASFs. Moreover, it is coherent with recent experimental studies demonstrating that HIF1 knockdown reduces glycolytic metabolism in human synovial fibroblasts [73].

## 4. Discussion

This work presents the construction of the first RASF-specific asynchronous Boolean model combining signaling, gene regulation, and metabolism. To obtain this model, the state-of-the-art RA-map V2 [26] was used along with the map-to-model framework proposed by Aghamiri et al. [36]. The model was inferred based on RASF-specific pathways and components according to scientific literature. Additional biocuration ensured high cellular specificity for RASFs. The model's RASF-specific behavior was validated against biological scenarios extracted from the scientific literature and based on small-scale functional experiments. The integration of this Boolean model with a constraint-based model of human central metabolism, MitoCore [41], allowed to assess the effects of RASFs' regulatory outcomes on their central metabolic flux distribution. The main strength of this approach lies in the reliability of both manually curated models rather than data-driven networks. It allows to address a lack of large-scale RASF-specific omics data in the regulatory network's construction as well as improper automatic reconstructions for the metabolic network.

To optimize the coupling of the two models, cell and disease-specific initial conditions were used to parametrize the regulatory model and decrease its complexity. The proposed framework only extracts constraints from metabolic compounds with a proven "inactive" asymptotic behavior, which allows it to automatically handle models with hundreds of components. This approach improves on previous attempts to couple Boolean models with metabolism, such as FlexFlux [33]. In FlexFlux the discrete qualitative states of the Boolean regulatory network are translated into several user-defined continuous intervals, while in this framework only the metabolic reaction fluxes whose regulatory components have maximal trap-spaces values equal to 0 are constrained. This choice is motivated by the difficulty of manually defining initial values and qualitative states to continuous intervals' equivalence for every component of large regulatory models. In addition, this framework adopts the asynchronous update as less deterministic and the identification of trap-spaces, using value propagation, to find states closer to the biological reality. This approach can facilitate the analysis of models with a higher number of inputs. However, finding meaningful combinations of initial conditions can be challenging for models including less-studied entities. Moreover, regulatory models presenting less metabolic-associated maximal values of trap-spaces equal to 0, would obviously have fewer constraints applied to metabolic fluxes, decreasing the number of alterations of metabolic fluxes shown by FBA.

The results of the hybrid RASF model's simulations under RASF-specific conditions revealed a highly glycolytic metabolism, along with a decreased oxidative metabolism for ATP production and confirmed a hypoxic environment around RASFs. Comparison with the control FBA's simulation results clearly demonstrated a metabolic reprogramming of RASFs. These results are consistent with recent experimental studies highlighting a glycolytic switch in RASFs [24,74].

Simulations of knock-outs and knock-ins of the regulatory initial conditions revealed HIF1 as a critical regulator of RASFs' metabolic reprogramming. HIF1 is a master transcriptional

factor involved in cellular and developmental response to hypoxia. Already identified in RA as a key effector of inflammation, angiogenesis, and cartilage destruction [75,76], HIF1 appears to be also involved in RASFs' metabolic alterations. It seems to promote glycolytic energy production by upregulating glucose transporters' expression (i.e. GLUT1 and GLUT3) and transcription of enzymes responsible for intracellular glucose breakdown through glycolysis (e.g. Hexokinase, Phosphofructokinase-1, Aldolase), similarly to observations in Cancer-Associated Fibroblasts (CAFs) [77,78]. In parallel, HIF1 might decrease ATP production through OXPHOS by transactivating genes responsible for $O_2$ demand and mitochondrial activity, such as Pyruvate Dehydrogenase Kinase 1 or MAX Interactor 1. Considered together, these various properties enable HIF1 to enhance glycolysis' rates as a crucial step of metabolic response to hypoxia [78].

RASFs' metabolic alterations are generally attributed to their stressful microenvironment but could also be considered from the perspective of metabolite exchange between RASFs and other RA joint's cells. Indeed, FBA results demonstrated several other altered metabolic pathways in RASFs, including fatty acids, amino acids, and reductive carboxylation. The latter observations have not yet been experimentally studied in RASFs but are similar to the ones made in CAFs. CAFs would undergo metabolic reprogramming to turn into "metabolic slaves", generating energy-rich fuels and nutrients to feed neighboring cancer cells and help sustain their aggressive activity [79]. Similarly, RASFs reside close to bio-energetically demanding cells (e.g. macrophages, dendritic cells, chondrocytes), experience a glycolytic switch and, according to these simulations, secrete high levels of energy-rich fuels and nutrients. Said nutrients are known to be involved in disease-associated behaviors and some experimental evidence suggests intracellular metabolic exchanges between RASFs and neighboring cells [80]. Thus, a reverse Warburg relationship may occur between RASFs and RA joint cells. RASFs would undergo a metabolic switch and reprogram their metabolism to adapt to their hypoxic environment and provide crucial metabolic intermediates to neighboring cells to sustain their inflammatory activity. Further experimental studies are needed to decipher the intricate mechanisms of these metabolic exchanges, the precise beneficiaries of such metabolic intermediates, and the primary signal for this metabolic reprogramming.

## 5. Perspectives

In this approach, a "linear" view of events is adopted, with signaling and gene regulation impacting metabolic pathways. However, the metabolic alterations due to HIF1's regulatory action may initiate and contribute to a hypoxic feedback loop. To have a more realistic overview of RASFs' dynamic behavior, further including the feedback of metabolism on gene regulation and signaling would be needed. An extension of this framework could be envisioned by adding a second timescale, as proposed by Thuillier et al. [81], with the regulatory network getting updated once the metabolic network is at steady state. This extension would allow for a complete view of the metabolic reprogramming. Finally, the metabolic analyses performed within the framework do not require particularly high computing power, making them suitable for larger metabolic networks. Coupling the Boolean regulatory model with a larger human genome-scale metabolic model could be considered to expand identification of altered metabolic pathways in RASFs.

## 6. Conclusion

Hybrid modeling methods become increasingly important, especially in the era of systems biology, where biological phenomena are recognized as resulting from complex interactions within different layers. In RA, computational models able to span multiple biological processes could help decipher the intricate mechanisms at the origin of its pathogenesis. This work

presents the first hybrid RASF model combining a RASF-specific asynchronous Boolean model with a global metabolic network. This hybrid RASF model is able to successfully reproduce the metabolic switch induced by signaling and gene regulation. Simulation results also enable further hypotheses on the potential reverse Warburg effect in RA. RASFs would reprogram their metabolism to produce more ATP, sustain their aggressive phenotype, and feed neighboring energetically demanding cells with fuels and nutrients. HIF1 was identified as the primary molecular switch driving RASFs' metabolic reprogramming. Already recognized as an important factor in the resolution of inflammation, targeting HIF1 might represent a promising path in the treatment of RA by addressing RASFs' metabolic reprogramming.

## Supporting information

**S1 Fig. Inference of the RASF model from the RA-map V2 (Zerrouk et al., 2022)** [26]**. (A)** Representation of the RA-map V2. The map is compartmentalized from top to bottom to represent the flow of information from the extracellular space through the plasma membrane, cytoplasm (including mitochondria and endoplasmic reticulum), nucleus, secreted compartment space and phenotypes. Proteins are represented in purple, complexes in pink, genes in dark green, RNAs in red, simple molecules in blue, metabolic enzymes in light green, and phenotypes in yellow. State transitions are represented by black arcs and inhibitions in red. **(B)** GINsim visualization of the RASF model inferred from the RA-map V2 by CaSQ. The layout is similar to the RA-map V2's layout but the compartments are not conserved. All components are represented as ellipsoid nodes. Green and red arcs denote positive and negative interactions.
(TIFF)

**S2 Fig. Visualization of the metabolic constraint extraction approach.** (A) RASF model's visual representation with metabolic pathways displayed in yellow. Regulatory components are not considered within the hybrid modeling framework to extract additional metabolic constraints. (B) Metabolic reactions associated with metabolic components presenting maximal associated trap- spaces equal to 0 are constrained to 0 in MitoCore. Remaining reactions are not considered within the hybrid modeling framework to extract additional metabolic constraints.
(TIFF)

**S1 Table. Common metabolic intermediates considered as metabolites in the MitoCore model.**
(XLSX)

**S2 Table. Biological scenarios identified from literature used for generic validation of the RASF model's behavior in the Cell Collective platform.** Experimental conditions were used as the model's inputs and its outputs were compared to the biological outcome to confirm or refute a scenario's validation.
(XLSX)

**S3 Table. RASF model's trap-spaces.** Metabolic components are marked with an asterisk.
(XLSX)

**S4 Table. Metabolic enzymes' projected maximal regulatory trap-space equal to 0 in RASF-specific initial conditions and their associated catalyzed reaction constrained to 0 in MitoCore.**
(XLSX)

**S5 Table. Metabolites' projected maximal regulatory trap-space equal to 0 in RASF-specific initial conditions and their associated producing reactions constrained to 0 in Mito-Core.**
(XLSX)

**S6 Table. Uptake reactions' C-fluxes in the control condition FBA with maximal ATP production as objective function.**
(XLSX)

**S7 Table. Secretion reactions' C-fluxes in the control condition FBA with maximal ATP production as objective function.**
(XLSX)

**S8 Table. Uptake reactions' C-fluxes in the RASF-specific condition FBA with maximal ATP production as objective function.**
(XLSX)

**S9 Table. Secretion reactions' C-fluxes in the RASF-specific condition FBA with maximal ATP production as objective function.**
(XLSX)

**S10 Table. Metabolic flux distribution in control conditions with maximal ATP production as objective function.**
(XLSX)

**S11 Table. Metabolic flux distribution in RASF-specific conditions with maximal ATP production as objective function.**
(XLSX)

## Author Contributions

**Conceptualization:** Anna Niarakis.

**Data curation:** Sahar Aghakhani, Anna Niarakis.

**Formal analysis:** Sahar Aghakhani, Sylvain Soliman.

**Funding acquisition:** Sylvain Soliman, Anna Niarakis.

**Investigation:** Sahar Aghakhani, Anna Niarakis.

**Methodology:** Sahar Aghakhani, Sylvain Soliman, Anna Niarakis.

**Project administration:** Sylvain Soliman, Anna Niarakis.

**Resources:** Sylvain Soliman, Anna Niarakis.

**Supervision:** Sylvain Soliman, Anna Niarakis.

**Validation:** Sahar Aghakhani, Sylvain Soliman, Anna Niarakis.

**Visualization:** Sahar Aghakhani.

**Writing – original draft:** Sahar Aghakhani, Anna Niarakis.

**Writing – review & editing:** Sahar Aghakhani, Sylvain Soliman, Anna Niarakis.

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
