## [Decision Letter · Decision Letter 0]

12 Sep 2022

Dear Dr. Niarakis,

Thank you very much for submitting your manuscript "Metabolic Reprogramming in Rheumatoid Arthritis Synovial Fibroblasts: a Hybrid Modeling Approach" for consideration at PLOS Computational Biology.

As with all papers reviewed by the journal, your manuscript was reviewed by members of the editorial board and by several independent reviewers. In light of the reviews (below this email), we would like to invite the resubmission of a significantly-revised version that takes into account the reviewers' comments.

We cannot make any decision about publication until we have seen the revised manuscript and your response to the reviewers' comments. Your revised manuscript is also likely to be sent to reviewers for further evaluation.

Sincerely,

James R. Faeder

Academic Editor

PLOS Computational Biology

Mark Alber

Section Editor

PLOS Computational Biology

Reviewer's Responses to Questions

**Comments to the Authors:**

Reviewer #1: In this manuscript, authors apply a combination of large-scale boolean models of signaling and gene regulation with flux balance analysis to deepen in the metabolic phenotype of Rheumatoid Arthritis Synovial Fibroblasts (RASFs). The manuscript is nicely written and structured, facilitating the reading and presenting the results in a coherent order. The proposed approach is appealing from the Systems Biology perspective, as authors attempt to understand the metabolic switch of RASFs combining available prior knowledge on the three main types of biological networks, namely signaling, gene-regulation and metabolism. Nevertheless, several of the assumptions made in this work need further support, and the proposed models present certain weaknesses that could be strengthened by addressing the points outlined below.

Major points:

- According to the first paragraph in the results section, the RA-map v2 contains 359 nodes, 14 inputs and 642 interactions that comprise signaling and gene regulatory interactions that are relevant in RA according to bibliographic evidence. These numbers fall short when compared to the whole number of gene expression changes that have been detected in previous analyses (check Lindberg et al. 2010). Authors should try to expand the underlying regulatory model content, and put it in the context of the total number of changes that have been detected in previous omics experiments.

- Similarly, the context-specificity of the model towards RASF biology is based on information from the same underlying map. Since this is one of the main claims of this paper, and given the dependence of other results on this assumption, authors need to provide additional evidence to support the cell-type specificity claim. Fortunately, there are two recent and extensive datasets where single-cell omics analyses were performed for synovial tissues in RA patients (Stephenson et al. 2018 and Zhang et al. 2019). Authors could use information from these datasets to support the initial conditions that they define and which theoretically transform the generic RA-map into a RASF specific map.

- On the same line, authors employed a genome-scale metabolic model which is small when compared to any comprehensive and recent genome-scale metabolic model of human cells. An example is the Human-GEM described in Robinson et al. 2020, which contains 13069 reactions (in comparison to the 324 + 83 included in MitoCore) and 8366 metabolites (in comparison to 74 included in MitoCore). Why do authors employ such a small metabolic model? How do they know that the many metabolic events that are neglected by using such a small metabolic model are not relevant for the metabolic switch occurring in RASF? Authors need to provide further justification in the selection of the GEM selected.

- Finally, authors need to demonstrate the advantages of their hybrid approach over other methods to achieve context-specific GEMs (see Hovratin et al. 2022). Using the single-cell information previously mentioned, authors can try to create a context specific GEM by setting an objective function that maximizes the overlap between enzymes that participate on selected metabolic reactions and genes specifically expressed in RASF obtained from the single-cell data analysis.

Minor points:

- Given that two major points include the expansion and the creation of new context-specific GEMs, it is expected that the solution space will grow significantly, turning the model into a non-identifiable model. In this context, authors can apply DEXOM, a recent tool to explore alternative solutions in context-specific GEMs (see Rodriguez-Mier et al 2021).

- A visualization of the trap states that occur during the regulatory model simulation would be beneficial for the understanding of the proposed approach.

- For the sake of reproducibility, authors should deposit all the code and data needed to reproduce their results in a permanent archive like Zenodo (https://zenodo.org/) or figshare (https://figshare.com/), not only in GitLab.

Reviewer #2: The authors present a hybrid modeling approach to analyse the rheumatoid arthristis synovial fibroblast (RASFs). They combine qualitative regulatory network with metabolic network and use the regulatory trap-spaces as constraints on the metabolic network. Their method is original and the integration of trap-spaces into FBA allows to propose a formal framework to the study the coupling of metabolic and regulatory networks. Their method allows to find expected experimental results such that the glycolytic switch.

The paper is suitable for the journal but there are some points which need to be explained:

1- The integration of regulatory constraints in constraint-based metabolic models is not an obvious task. Recent works have begun to take into account both networks. The authors discuss about why they do not choice to use existing tools like flexflux (l.555) which is motivated by the difficulty to define initial conditions and intervals in flexflux. The authors propose to use trap-spaces to overcome these difficulties. However, the reader would like to be convinced about the biological interest of trap-spaces in the metabolism. Could the authors explain more about it ?

2- The trap-spaces are stable motifs which the system can not escape. However some asymptotic behaviors oscillate. How the authors manage with the oscillatory behaviors of the regulatory networks ?

3- The authors have to explain "minimal" trap-spaces line 251 ? what it means to calculate trap-spaces without performing simulation (line 257) ? Could they discuss about the computation time and the enumeration of trap-spaces ?

4- They show 8 Trap-spaces in the supplementary information but the reader has difficulty understanding their biological meanings. Do they represent a specific phenotype profile ? what differentiates them ?

5- The figure 5 shows the main active pathways of the ccm however, the authors test 14 initial conditions. How were these conditions chosen? In all trap-spaces of the RASF model, the PDHm = 0 which necessary prevent to use the OXPHOS and thus lead a glycolytic profile to produce ATP. Could the authors discuss their initial choice and the link with the enzyme trap-spaces ?

6- Figure 4B is not readable, what should we understand ?

**Have the authors made all data and (if applicable) computational code underlying the findings in their manuscript fully available?**

Reviewer #1: **No: **For the sake of reproducibility, authors should deposit all the code and data needed to reproduce their results in a permanent archive like Zenodo (https://zenodo.org/) or figshare (https://figshare.com/), not only in GitLab.

Reviewer #2: Yes

PLOS authors have the option to publish the peer review history of their article (what does this mean?). If published, this will include your full peer review and any attached files.

Reviewer #1: No

Reviewer #2: No
---

## [Decision Letter · Decision Letter 1]

11 Nov 2022

Dear Dr. Niarakis,

We are pleased to inform you that your manuscript 'Metabolic Reprogramming in Rheumatoid Arthritis Synovial Fibroblasts: a Hybrid Modeling Approach' has been provisionally accepted for publication in PLOS Computational Biology.

Best regards,

James R. Faeder

Academic Editor

PLOS Computational Biology

Mark Alber

Section Editor

PLOS Computational Biology

Reviewer's Responses to Questions

**Comments to the Authors:**

Reviewer #1: The authors responded satisfactorily to all my points. Most of my comments were aimed at extending and validation the underlying modeling strategy. The authors explained that the underlying prior knowledge already borrows information from omics data (Zerrouk et al. 2022). They also justified their choice of metabolic model size by claiming that, in the trade-off of confidence versus coverage, they chose to stay with a smaller model that they can rely on more.

**Have the authors made all data and (if applicable) computational code underlying the findings in their manuscript fully available?**

Reviewer #1: Yes

PLOS authors have the option to publish the peer review history of their article (what does this mean?). If published, this will include your full peer review and any attached files.

Reviewer #1: No

---

## [Editor Report · Acceptance letter]

7 Dec 2022

PCOMPBIOL-D-22-01097R1 

Metabolic Reprogramming in Rheumatoid Arthritis Synovial Fibroblasts: a Hybrid Modeling Approach

Dear Dr Niarakis,

I am pleased to inform you that your manuscript has been formally accepted for publication in PLOS Computational Biology. Your manuscript is now with our production department and you will be notified of the publication date in due course.

With kind regards,

Anita Estes
